# Effects of Short-Term Treatment with α-Lipoic Acid on Neuropathic Pain and Biomarkers of DNA Damage in Patients with Diabetes Mellitus

**DOI:** 10.3390/ph17111538

**Published:** 2024-11-16

**Authors:** Juozas R. Lazutka, Kristina Daniūnaitė, Veronika Dedonytė, Aistė Popandopula, Karolina Žukaitė, Žydrūnė Visockienė, Laura Šiaulienė

**Affiliations:** 1Life Sciences Center, Vilnius University, Saulėtekio Al. 7, LT-10257 Vilnius, Lithuania; kristina.daniunaite@gf.vu.lt (K.D.); veronika.dedonyte@gmc.vu.lt (V.D.); aiste.popandopula@gmc.stud.vu.lt (A.P.); karolina.zukaite@mf.stud.vu.lt (K.Ž.); 2Faculty of Medicine, Vilnius University, M. K. Čiurlionio St. 21, LT-03101 Vilnius, Lithuania; zydrune.visockiene@santa.lt; 3Vilnius University Hospital Santaros Klinikos, Santariškių St. 2, LT-08661 Vilnius, Lithuania

**Keywords:** neuropathic pain, α-lipoic acid, diabetic polyneuropathy, DNA damage

## Abstract

Background/Objectives: Diabetes mellitus (DM) is a complex and heterogenous disease classified as a group of metabolic disorders characterized by chronic hyperglycemia resulting from defects in insulin secretion, insulin action, or both. It leads to various complications, some of which are macrovascular or microvascular complications, like diabetic polyneuropathy (DPN), having a profound impact on patients’ quality of life. Oxidative stress (OS) is one of the significant mechanisms in the development and progression of DPN. Thus, targeting OS pathways by antioxidants, such as α-lipoic acid (ALA), could represent a promising therapeutic strategy for alleviating neuropathic symptoms. The aim of our study was to evaluate whether short-term (from 4 to 9 days) intravenous administration of ALA could cause any measurable improvement in subjects with DM. Methods: Sixteen subjects with DM (six type 1 and ten type 2) and sixteen nondiabetic subjects matched by sex and age were recruited to this study. Only subjects with DM received treatment with ALA (600 mg daily). Pain intensity and biomarkers of DNA damage including plasma concentration of 8-hydroxy-2′-deoxyguanosine (8-OHdG), frequency of micronucleated lymphocytes (MN), and frequency of sister-chromatid exchanges (SCEs), were measured before and after the treatment with ALA. Results: Pain intensity and 8-OHdG levels were significantly lower in DM subjects after the ALA treatment than before the treatment. However, no changes in the frequency of SCEs and MN were observed. Conclusions: Our results show some evidence that even a short-term intravenous treatment with ALA could be beneficial for diabetic subjects, reducing pain intensity and concentration of 8-OHdG in blood plasma.

## 1. Introduction

Diabetes mellitus (DM) is a complex and heterogeneous disease classified as a group of metabolic disorders characterized by chronic hyperglycemia resulting from defects in insulin secretion, insulin action, or both. Type 1 (T1DM) and type 2 (T2DM) are the most prevalent DM forms, and it is now well recognized that these two types are distinct disorders with certain aspects of etiology, pathogenesis, clinical presentation, and complications. Both types manifest with sustained hyperglycemia, however, T1DM is caused by autoimmune injury to pancreatic beta cells producing insulin, while T2DM develops due to insulin resistance caused by different metabolic risk factors such as obesity and dyslipidemia [1]. Diabetic polyneuropathy (DPN) is a common microvascular diabetic complication with up to 34% estimated lifetime prevalence among subjects with T1DM and over 50% lifetime prevalence among subjects with T2DM. Importantly, the prevalence of DPN is low in subjects with newly diagnosed T1DM and relatively high (may reach up to 30%) in newly diagnosed and early T2DM subjects. Symptomatic and painful DPN accounts for 50% and 30% of all DPN cases, respectively [2]. Painful DPN is characterized by hyperalgesia, allodynia, and pain, which could be described as burning, stabbing, cramping, or shooting sensations that are usually worse at night. This condition has a profound impact on patients’ quality of life, resulting in sleep disruptions, decreased capacity for daily activities, as well as increased morbidity and mortality rates in individuals with DM [2].

In both T1DM and T2DM patients, hyperglycemia is a common risk factor for DPN, however, insulin resistance, obesity, and dyslipidemia play a significant role in T2DM patients as well. Hyperglycemia and altered insulin signaling in T1DM, along with hyperglycemia, insulin resistance, and dyslipidemia in T2DM, result in dysfunction of the endoplasmic reticulum and mitochondria, systemic inflammation, and oxidative stress (OS). These events, in turn, lead to the activation of the polyol, hexosamine, protein kinase C, and advanced glycation end products (AGEs) pathways that are detrimental to microvascular structures. Additionally, the expression of multiple molecules is upregulated, including, among others, cytokines/chemokines, various kinases, and the NF-κB family of redox-sensitive transcriptional regulators [3,4].

Numerous studies indicated that OS is one of the most significant mechanisms in the development and progression of DPN in both T1DM and T2DM subjects (reviewed in [5]). Oxidative stress is caused by an imbalance between the generation and detoxification of reactive oxygen species (ROS) in cells and tissues [3]. Under normal conditions, ROS are involved in various cellular and physiological processes such as cell signaling, immune response, regulation of gene expression, and maintenance of cellular homeostasis [6,7,8]. However, when the formation of ROS and the capability to clear them is not balanced, OS may occur. OS has multifaceted effects in the pathogenesis of DPN, primarily causing damage to nerve cell structures along with alterations to lipids, proteins, and DNA. Indeed, increased levels of 8-hydroxy-2′-guanosine (8-OHdG), the main product of DNA oxidation, were associated with microvascular complications and increased mortality in diabetic subjects [9,10]. In addition, other studies indicated increased genomic damage in diabetes, as measured by different cytogenetic endpoints such as chromosome aberrations, sister-chromatid exchanges (SCEs), and micronuclei (MN) [11,12,13,14]. Even more of this damage could be seen in diabetic subjects with different microvascular complications [15,16,17].

Management of DPN includes lifestyle modifications, diabetes therapy aimed at achieving near-normoglycemia, symptomatic pain relief, and pathogenesis-oriented pharmacotherapy [4]. Glycemic control has proven to be an effective measure for managing DPN in patients with T1DM, while lifestyle modifications are primarily effective for patients with T2DM [18]. Analgesic pain relief has limited efficacy and can cause adverse effects [4]. Therefore, pathogenesis-oriented therapy is gaining more attention in clinical practice. One of the pathogenetic targets for such therapy could be OS. Its harmful effects could be ameliorated by different antioxidant molecules, of both endogenous and exogenous origin [19]. These molecules include enzymatic antioxidants (e.g., superoxide dismutase, catalase, glutathione peroxidase), nonenzymatic antioxidants (vitamins C and E, flavonoids, carotenoids, etc.), and exogenous antioxidants (specific diets or food supplements). Targeting OS pathways with these antioxidants could represent a promising therapeutic strategy for mitigating cellular damage of nerve cells and alleviating neuropathic symptoms. One such nonenzymatic antioxidant molecule is α-lipoic acid (1,2-dithiolane-3-pentanoic acid, ALA), which stands out as the sole antioxidant demonstrating clinical efficacy and is recommended for DPN treatment in both T1DM and T2DM subjects [20]. In the human body, the main sources of ALA are endogenous de novo synthesis and intake of ALA-rich food (e.g., red meat, beets, carrots, potatoes, spinach, and broccoli). ALA is a cofactor of mitochondrial enzyme complexes involved in energy metabolism [21]. In addition, ALA has the ability to protect cells from the damaging effects of ROS by reducing the oxidized forms of other antioxidants [21,22]. ALA also enhances insulin sensitivity and shows the potential in preventing microvascular complications [23,24] and ameliorating comorbidities like obesity, hypertension, and dyslipidemia [25]. Acting as a direct biological antioxidant and metal chelator, ALA scavenges ROS and reactive nitrogen species while indirectly regenerating endogenous antioxidants [26]. In addition, ALA is capable of reducing numbers of SCEs and MN induced by DNA-damaging substances in human lymphocytes in vitro [27] and rat bone marrow cells in vivo [28].

During the last decades, different randomized controlled clinical studies showed promising results of ALA therapy on neuropathic symptoms and impairments [4,29], with the response rate of about 40–60% [30]. However, due to the pharmacokinetic properties of ALA (short plasma half-time), the intravenous route of treatment seems to be more efficacious than the oral one [31]. Moreover, the most efficient, compared to placebo, are intravenous infusions of ALA (600 mg/day) for three weeks or longer periods lasting up to six months [4]. Such intravenous treatment is extremely inconvenient for both patients and their physicians because it needs long-term hospitalization or everyday visits to outpatient departments. For this reason, in current clinical practice, ALA is usually administered intravenously for a short-term period, for two or more weeks, then followed by a long-term oral administration [32]. However, even such intravenous treatment duration might be inconvenient for most patients.

Conditions under which many randomized clinical trials are conducted often differ from the realities of clinical practice [33]. For example, in real-world situations neither physicians nor patients have the possibility to receive intravenous ALA treatment for weeks or even months—as mentioned before, the usual practice is to deliver this drug for a few days or, in the best case, a few weeks. As a response to such differences between conditions of randomized clinical trials and usual clinical practices, a growing number of studies based on real-world situations are conducted [34]. The results of these studies could be considered as complementary to the findings of clinical studies.

The aim of our study was to evaluate if short-term (less than 10 days) intravenous administration of ALA used under real-life conditions could cause any measurable improvement in subjects with DM. As a subjective index, we chose pain intensity measured using the Universal Pain Assessment Tool (UPAT) [35]. We also used three objectively measured biomarkers of DNA damage that could be caused by oxidative stress—concentration of 8-OHdG in blood plasma, frequency of MN in lymphocytes, and frequency of SCEs in chromosomes of lymphocytes obtained from DM subjects before and after the treatment with ALA. 8-OHdG is a product of DNA oxidation by reactive oxygen species. MN represents small additional nuclei formed from broken parts or whole chromosomes lost during mitosis and is a standard measure of DNA damage [36]. In addition, higher frequency of MN has been shown to be associated with increased cancer risk in human populations [37]. SCEs correlate well with DNA damage, repair, or its effect on replication and represent symmetrical exchanges between two sister chromatids of the same chromosome occurring by the mechanism of homologous recombination [38]. Increased SCE frequency could indicate genomic instability and is also observed in some cancers [39]. As mentioned above, these three biomarkers of DNA damage were also associated with diabetes and its microvascular complications [9,10,11,12,13,14,15,16,17].

Our results showed some evidence that even a short-term intravenous treatment with ALA could be beneficial for diabetic subjects, reducing pain intensity and concentration of 8-OHdG in blood plasma. However, no changes in the frequency of SCEs and MN were observed.

## 2. Results

Due to technical reasons, not all biomarkers of DNA damage were evaluated for all subjects. 8-OHdG levels were measured in blood plasma of 14 control subjects, 14 DM subjects before ALA treatment, and 9 DM subjects after ALA treatment. MN were analyzed for 14 DM subjects, and SCEs were analyzed for all 16 DM subjects. The results of all analyses, together with some demographic and clinical characteristics of all subjects can be found in Appendix A.

The results of pain intensity measurements in DM subjects before and after treatment with ALA are shown in Figure 1. For three subjects, initial pain intensity was zero, so it is obvious that no pain intensity changes were observed after ALA treatment. For another subject, the initial pain level was 3, and it was retained after ALA treatment. For the rest 12 subjects, pain intensity dropped after ALA treatment, for some of them quite drastically (for example, from 10 to 5, from 8 to 2, or from 6 to 0), for others—moderately (for example, from 7 to 6 or from 4 to 2). Overall, pain intensity was significantly lower in DM subjects after the ALA treatment than before the treatment (*p* = 0.0024, effects size 0.88, Paired Wilcoxon Signed Rank test). There was no difference in mean pain intensity between T1DM and T2DM subjects before (6.0 vs. 7.0, *p* = 0.6953, Mann–Whitney U-test) and after the treatment with ALA (2.5 vs. 3.9, *p* = 0.0696). For both groups, pain intensity was significantly lower after ALA therapy (*p* = 0.0265 for T1DM subjects, and *p* = 0.0049 for T2DM subjects, Mann–Whitney U-test). Mean pain level before ALA treatment did not differ in males and females (6.0 vs. 7.1, respectively), the same was true after ALA treatment (3.4 vs. 3.5).

DM subjects before ALA treatment had slightly higher 8-OHdG concentrations in their blood plasma, mainly because of two outliers with high 8-OHdG levels (Figure 2a). After the treatment with ALA, levels of 8-OHdG were significantly lower (Figure 2b; *p* = 0.0039, Paired Wilcoxon Signed Rank test). It is important to note that 8-OHdG levels after ALA treatment were to some extent lower for all nine subjects, for whom 8-OHdG tests were available (effect size 0.96). Since 8-OHdG plasma concentrations after ALA treatment were measured only in three T1DM subjects (Appendix A), performing any meaningful statistical analysis in separate groups of T1DM and T2DM subjects was impossible.

The results of the analysis of SCEs and MN are shown in Figure 3. DM subjects had a higher frequency of SCEs (*p* = 0.0075, Mann–Whitney U test) but not MN (*p* = 0.4232) than controls. There were no statistically significant differences in mean SCE and MN values between T1DM and T2DM subgroups. Frequency of SCEs positively correlated with pain intensity before (Spearman’s rank correlation coefficient r_s_ = 0.6543, *p* = 0.006) but not after ALA treatment (r_s_ = 0.0638, *p* = 0.8145). No statistically significant changes in the frequency of both cytogenetic biomarkers of DNA damage were observed after the treatment with ALA.

## 3. Discussion

The main goal of our study was to evaluate if short-term administration of ALA used under real-life conditions could improve some oxidative stress-related indices in patients with T1DM and T2DM. We found that intravenous treatment with ALA 600 mg/day for 4–9 days could be beneficial for subjects with diabetes, reducing their pain intensity and concentration of 8-OHdG in blood plasma.

Effects of ALA treatment on diabetic polyneuropathy have been studied in numerous studies, both observational and clinical trials. While there is general agreement that ALA treatment has favorable effects on many symptoms of DPN [4,29] and improves overall quality of life [40], some controversies still exist regarding the effectivity of ALA treatment on neuropathic pain. For example, one recent meta-analysis [41] concluded that ALA treatment did not reduce pain in diabetic patients, while another recent meta-analysis [42] came to the opposite conclusion that ALA could reduce pain symptoms. Such discrepancies may be due to a mode of treatment with ALA since it has been shown that intravenous infusions are more effective in neuropathic pain treatment than oral administration [43]. Indeed, pharmacokinetic studies have shown that the oral bioavailability of ALA is only 30% [44]. The plasma half-life of ALA is about 0.5–1.5 h, depending on the formulation [45], and there is no evidence of its accumulation in the body [46]. After intravenous treatment with ALA, the plasma half-life remained almost the same; however, much higher peak plasma concentrations were achieved [47]. Some researchers have speculated that the higher effectiveness of intravenous ALA administration is due to the higher plasma concentration of the drug, which is necessary for glutathione regeneration in target cells, for example [31,45]. Since in our study ALA was infused intravenously, our results support findings about the effectiveness of intravenous administration of ALA in reducing neuropathic pain.

A recent study obtained results very similar to ours (reduction in pain level by ~3.5 points) with only slightly longer (for 14 days) intravenous treatment with ALA [48]. In this study, pain level was assessed using Visual Analog Scale (VAS), and three therapeutic cycles were performed. Therapeutic cycle lasted for 14 days, followed by a pause for 6–7 weeks. It is interesting to note, that pain levels increased after the pause (though to a level lower than the initial one), and then statistically significantly decreased again after intravenous infusion of ALA. Based on these data and our results, and considering that previous clinical studies have shown no significant advantage of long-term (6 months to 4 years) treatment schedules over short-term (3–5 weeks) treatment schedules [18], repeated short-term intravenous treatment (1–2 weeks) with ALA could be a promising strategy for the treatment of neuropathic pain.

In addition, ALA therapy could be useful in other painful conditions because it was shown to be effective in the treatment of radicular pain [49], migraine, carpal tunnel syndrome, burning mouth syndrome [42], and pain with unknown etiology [50]. However, no positive effects of ALA treatment were observed in patients with fibromyalgia, including diabetic ones [51]. In addition, ALA efficiency in treating different pain forms has been shown in numerous animal studies [44]. Molecular mechanisms of ALA action on neuropathic pain have been studied quite extensively as well. It has been shown that ALA reduces neuropathic pain by scavenging ROS, inhibiting transcriptional factors involved in immune response, regenerating other antioxidants like vitamins C and E and glutathione, metal chelation, suppression of microglial activation, and preventing mitochondrial damage [44,52,53,54]. Overall, keeping all these data in mind, we believe that intravenous ALA treatment, even for a short time, could be used for the management of neuropathic pain.

In our study, we observed significantly lowered concentrations of 8-OHdG in the blood plasma of subjects with diabetes after the treatment with ALA. 8-OHdG is a well-known biomarker of oxidative stress in both the plasma [9,10] and urine [55] of patients with diabetes. In several previous studies, different outcomes of ALA treatment on the concentration of 8-OHdG have been reported. For example, in some studies, decreased serum [56] or urine [57] concentrations of 8-OHdG after ALA treatment were found, while other studies failed to find significant changes in concentration of 8-OHdG [58] or other markers of oxidative stress [59]. It is important, however, that lowered concentrations of 8-OHdG after ALA treatment were found in different issues of laboratory animals [60,61]. Poulsen et al. [62] suggested that a plasma level of 8-OHdG is not a good measure of oxidative stress because it mainly depends on the patient’s kidney function. However, since all our study patients had normal kidney function before treatment and were administered ALA only for 4–9 days, we did not expect any significant changes in kidney function in such a short time period. Under such circumstances, lowered plasma concentrations of 8-OHdG could reflect real changes in oxidative stress levels [62].

We also studied two cytogenetic biomarkers of DNA damage—frequency of micronuclei (MN) in lymphocytes and of sister-chromatid exchanges (SCE) in chromosomes of patients with diabetes and control subjects. As in some other studies [63,64,65], we found an increased frequency of SCEs in patients with diabetes as compared to control, but no significant differences in MN frequency. Intravenous infusions with ALA did not significantly change the frequency of either of these biomarkers. These results are not unexpected because both MN and SCEs reflect DNA damage accumulated in lymphocytes during their whole lifetime. Since blood samples for cell cultures were drawn immediately after the last dose of ALA, perhaps there was not enough time for any significant changes in MN and SCE frequency. On the other hand, our study sample was small, and statistical power might be too low to detect small changes in MN and SCE frequency after ALA treatment, if any.

Our study, of course, has some limitations. It was designed not as a clinical trial but as a pilot observational study dealing with real-world situations. For this reason, we had neither control nor placebo groups for ALA treatment. However, our main goal was to test whether ALA is still beneficial for patients when it is intravenously administered for only a short period of time. Another limitation of our study is a small sample size that obviously influenced our results on MN and SCE frequency. However, in pairwise comparisons of pain and 8-OHdG before and after ALA treatment, we observed a very large effect size—0.88 and 0.96, respectively. We believe that such a large effect size supports the reliability of our findings despite the small sample size. Finally, our study group consists of both T1DM and T2DM subjects. Recent studies indicated that pathogenetic mechanisms of DPN and response to treatment may be different in patients with different types of DM [66]. Thus, in the future, the effects of short-term ALA treatment should be investigated in separate groups of T1DM and T2DM patients.

In conclusion, our results show some evidence that even short-term intravenous treatment with ALA could be beneficial for patients with diabetes and polyneuropathy, especially for alleviating neuropathic pain and diminishing oxidative stress. These results could be a good starting point for a larger randomized placebo-controlled study investigating the usefulness of short-term ALA treatment for neuropathic pain management.

## 4. Materials and Methods

### 4.1. Study Subjects

The study was conducted at Vilnius University Hospital Santaros Klinikos according to the Declaration of Helsinki and was approved by Vilnius Regional Biomedical Research Ethics Committee (registry number 2019/6-1146-635). All participants gave written consent. For a total, 6 subjects with T1DM, 10 with T2DM, and 16 control subjects without DM were enrolled in the study. Patients and controls were roughly matched by age and gender. Average disease duration was 15.7 years for T2DM patients and 13.7 years for T1DM. T2DM patients had more comorbid conditions than T1DM patients and controls. In all three groups, the most common comorbid conditions were ischemic heart disease, primary arterial hypertension, and dyslipidemia. None of the subjects were diagnosed with cancer. Number of chronic comorbidities was calculated exactly as described in Chima et al. [67]. Demographic and clinical characteristics of all study subjects are shown in Appendix A.

Diagnosis of diabetes was established according to medical records or, for subjects with newly diagnosed diabetes, according to World Health Organization diagnostic criteria [68]. Diabetic polyneuropathy (DPN) was diagnosed using the Neuropathy Symptom Score (NSS) as described previously [69], and neuropathic pain intensity was assessed by the Universal Pain Assessment Tool (UPAT) [35], which ranges from 0 (no pain) to 10 (‘pain as bad as it could possibly be’).

All but two diabetic subjects had glycated hemoglobin (HbA1c) levels > 7% mmol/L (Appendix A). The control group underwent morning fasting venous plasma glucose (FPG) testing showing that 13 subjects had FPG less than 5.6 mmol/L and three subjects had FPG in the range of 5.7–6.0 mmol/L (Appendix A).

Subjects with DM received an intravenous infusion of α-lipoic acid (Thiogamma Turbo Set 12 mg/mL, Wörwag Pharma GmbH & Co., KG, Böblingen, Germany) 600 mg daily from 4 to 9 days (Appendix A). Blood samples for the analysis were collected via venipuncture the morning before treatment and several hours after the last infusion.

### 4.2. Analysis of Biomarkers of DNA Damage

The DNA damage level in plasma samples was assessed using the DNA Damage Competitive ELISA kit following the manufacturer’s protocol (Invitrogen™, Thermo Fisher Scientific, Frederick, MD, USA). Although this kit detects mainly 8-hydroxy-2′-deoxyguanosine (8-OHdG), a minor fraction of other oxidized guanine species, 8-hydroxyguanosine and 8-hydroxyguanine, could be detected as well. A standard curve was generated in each experiment using the provided 8-OHdG solution, with 2× dilutions spanning the range of 0–8000 pg/mL. 8-OHdG levels were measured in 50 μL of plasma samples, diluted 1:8 with 1× Assay Buffer, in duplicate. Absorbance readings were taken at 450 nm using the Infinite M200 microplate reader (Tecan, Mannedorf, Switzerland) within 10 min of sample preparation.

For sister-chromatid exchange (SCE) analysis, heparinized whole blood samples were diluted at a ratio of 1:15 with RPM1 1640 media supplemented with 12% heat-inactivated newborn calf serum, 7.8 μg/mL of phytohemagglutinin, 10 μg/mL of 5-bromo-2′-deoxyuridine (BrdUrd), and 50 μg/mL of gentamicin. BrdUrd was used for differential labeling of sister chromatids that could be visualized after the two rounds of replication. All reagents were purchased from Sigma (St. Louis, MO, USA). Cells were cultured in sterile bottles for 72 h at 37 °C in 5% CO_2_ atmosphere. Colchicine was added into the culture for the last 3 h at a final concentration of 0.6 μg/mL. Cultures were harvested with centrifugation, followed by 30 min hypotonic treatment (0.075M KCl at 37 °C) and three periods of fixation in methanol–glacial acetic acid (3:1). Flame-dried slides were prepared and stained for 10 min with 10 μg/mL of Hoechst 33,258 dye (dissolved in 0.07 M Sorensen’s buffer, pH 6.8). Then, the slides were rinsed, mounted with citrate buffer (pH 8.5), covered with coverslips, and exposed to UV light (400 W mercury lamp at a distance of 15 cm) for 6–7 min. Slides were then rinsed and stained for 3–4 min with 5% Giemsa. These staining procedures allowed differential staining of sister chromatids in cells that underwent two replication cycles in the cell culture (Figure 4a). SCEs were counted as exchange sites between lightly and darkly stained chromatids of the same chromosome.

For the analysis of micronuclei (MN), cell culture was set up exactly the same as for SCE analysis, except that BrdUrd was not added to the culture mixture. Forty-four hours after the initiation of culture, cytochalasin B was added at a final concentration of 6 μg/mL in order to block cytokinesis and to obtain binucleated cells. After incubation for another 28 h (72 h in total), cells were harvested by centrifugation for 8 min at 400× *g*. Before fixation, cells were treated with the cold hypotonic solution (0.075 M KCl), then centrifuged immediately and fixed three times with cold fixative (methanol:acetic acid, 5:1; the first portion of fixative was diluted with an equal volume of 0.9% NaCl). The fixed cells were dropped onto slides and air-dried. Conventional staining with Giemsa stain was used. Micronuclei were identified as small additional nuclei in cytoplasm of cells that underwent cellular division in the cell culture (Figure 4b).

All slides were analyzed with a Nikon Eclipse E200 microscope (Nikon, Tokyo, Japan). Per each individual, fifty 2nd division metaphases were analyzed for SCEs, and no less than a thousand cytochalasin B-blocked binucleated cells were analyzed for MN. The standard scoring criteria for MN were used [36].

Statistical calculations were performed using the online statistical calculator Statistics Kingdom [70]. Comparison between two groups was made using the Mann–Whitney U-test, and for comparison of dependent samples, a paired Wilcoxon signed-rank test was used.

## Figures and Tables

**Figure 1 pharmaceuticals-17-01538-f001:**
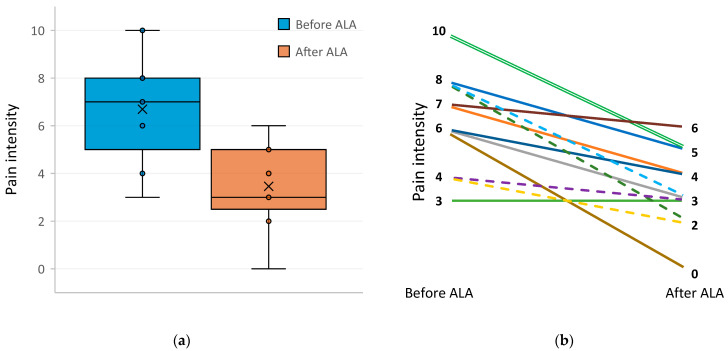
Pain intensity in subjects with DM before and after the therapy with α-lipoic acid (ALA), 600 mg daily from 4 to 9 days. (**a**) Box-and-whisker plots of neuropathic pain intensity assessed by the Universal Pain Assessment Tool (UPAT) [33] in DM subjects before and after the treatment with ALA; (**b**) Individual changes in pain intensity in DM subjects after the therapy with ALA; broken lines indicate T1DM subjects, solid lines—T2DM subjects; three subjects had pain intensity of zero before and after ALA therapy (two T1DM and one T2DM), therefore, they were not included into the graph; two subjects had pain intensity of 10 before the therapy and 5 after the therapy (shown by double line).

**Figure 2 pharmaceuticals-17-01538-f002:**
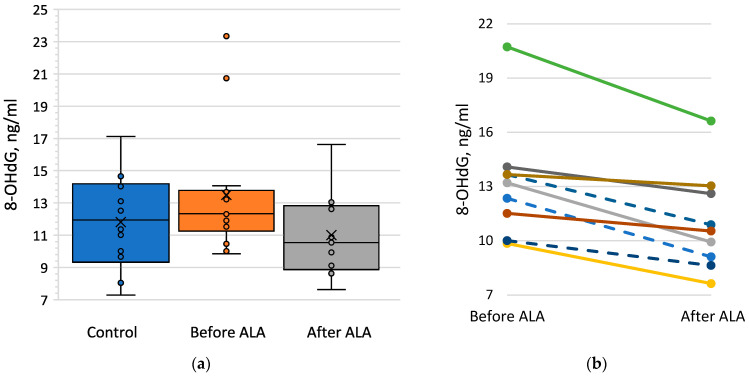
Effects of diabetes mellitus and treatment with α-lipoic acid (ALA) on 8-hydroxy-2′-deoxyguanosine (8-OHdG) blood plasma concentrations. (**a**) Box-and-whisker plots of 8-OHdG concentration in blood plasma of control subjects, DM subjects before the treatment with ALA, and DM subjects after the treatment with ALA; (**b**) Individual changes of 8-OHdG concentrations in blood plasma of nine DM subjects for whom 8-OHdG tests were performed both before and after the therapy with ALA; broken lines indicate T1DM subjects, solid lines—T2DM subjects.

**Figure 3 pharmaceuticals-17-01538-f003:**
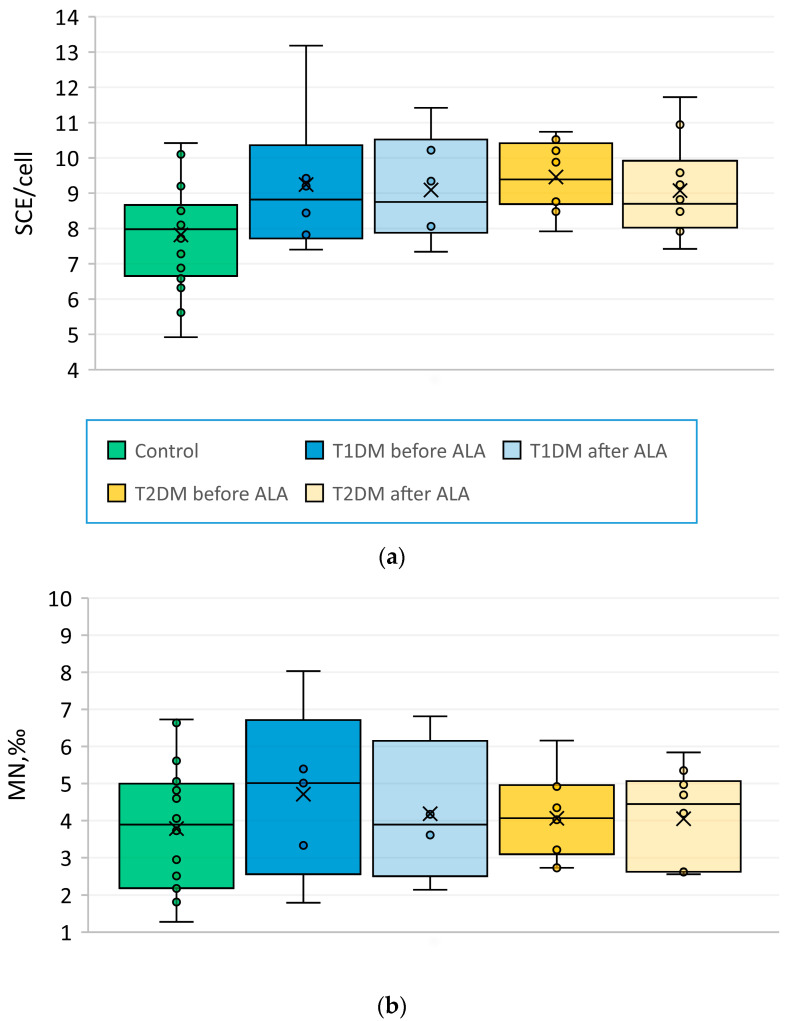
Effects of diabetes mellitus and treatment with α-lipoic acid (ALA) on cytogenetic biomarkers of DNA damage. (**a**) Box-and-whisker plots of sister-chromatid exchange (SCE) frequency in control subjects, T1DM and T2DM subjects before and after the treatment with ALA. (**b**) Box-and-whisker plots of micronuclei (MN) frequency in control subjects, T1DM and T2DM subjects before and after the treatment with ALA.

**Figure 4 pharmaceuticals-17-01538-f004:**
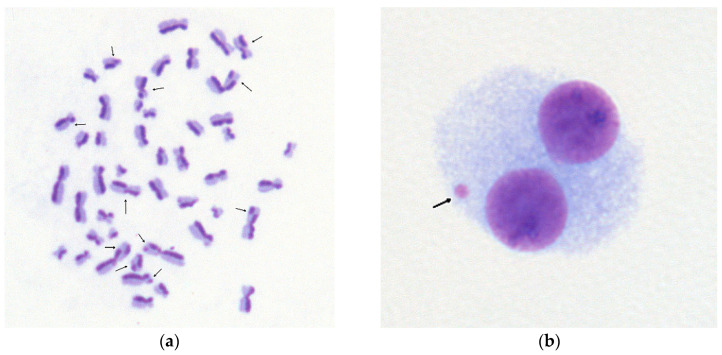
Sister chromatid exchanges (SCEs) and micronucleus (MN) in human peripheral blood lymphocytes. (**a**) Representative microphotograph of differentially stained human chromosomes obtained from cells grown in vitro for two cell cycles in the presence of 5-bromo-2′-deoxyuridine; SCEs are indicated by arrows. (**b**) Representative microphotograph of binucleated human lymphocyte; micronucleus—broken or lagging chromosome—is indicated by arrow.

## Data Availability

Data are contained within the article or Appendix A.

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
