# Peer review of "Effects of Short-Term Treatment with α-Lipoic Acid on Neuropathic Pain and Biomarkers of DNA Damage in Patients with Diabetes Mellitus"

_pharmaceuticals, 2024, doi:10.3390/ph17111538_

Round 1
Reviewer 1 Report
Comments and Suggestions for Authors
-
Small Sample Size: The study involved only 16 diabetic patients, which is a small sample size to draw strong, generalizable conclusions. This limitation is also mentioned regarding the analysis of biomarkers like 8-OHdG (only nine subjects analyzed post-treatment).
-
Limited Duration of Treatment: The study used a short-term treatment duration (4-9 days), which may not be sufficient to observe long-term effects of α-lipoic acid (ALA) on DNA damage markers like SCEs and MN. This could explain the lack of significant changes in these markers.
-
No Significant Changes in Cytogenetic Markers (MN and SCE): Despite the reduction in pain intensity and 8-OHdG levels, the study did not observe significant changes in MN and SCE frequencies. This is crucial as it raises concerns about whether ALA is truly effective at reducing DNA damage over such a short treatment window.
-
Lack of Control Group for ALA Treatment: There was no control group to compare the effect of ALA treatment in diabetic patients. Including a placebo group could help clarify the specific impact of ALA compared to other treatments or no treatment.
-
Discussion of Conflicting Literature: The authors mention contradictory results from meta-analyses regarding ALA's effectiveness in reducing neuropathic pain. This conflict should be discussed more comprehensively to explain how the present study fits into the broader literature.
-
Data Presentation (Figures):
- Figure 1: The representation of pain intensity in subjects before and after treatment is clear, but it is suggested to use error bars or standard deviations to improve visual interpretation of variability across the population.
- Figure 3: The microphotographs included in the figure (e.g., SCEs and MN) need higher resolution or improved contrast to make them easier to interpret. In their current form, they are difficult to analyze, especially for readers unfamiliar with these techniques.
-
Methodological Detail: The study briefly mentions the biomarkers and techniques (e.g., ELISA, microscopy), but more detailed protocols should be provided for reproducibility, particularly regarding the quantification of biomarkers such as SCEs and MN.
-
Inconsistency in Results Reporting: The results section does not consistently report the demographic and clinical characteristics of the study subjects, especially considering the gender and age of the participants. Demographic information is critical when interpreting pain perception and treatment effects.
Fatal Issues:
-
Statistical Power and Sample Size: Given the small sample size, the study lacks sufficient statistical power to detect minor differences, especially in DNA damage markers. This is a serious limitation that affects the reliability of the conclusions drawn, particularly in terms of generalizing the findings.
-
Lack of Long-Term Follow-up: The short duration of the study (4-9 days) is insufficient for determining the long-term effectiveness of ALA in treating diabetic neuropathy and its impact on biomarkers of DNA damage. Future studies should consider a longer treatment and follow-up period to capture the full extent of ALA’s effects.
-
Absence of Placebo-Controlled Design: A placebo-controlled design is essential in clinical trials to eliminate bias. The lack of such a group makes it difficult to attribute observed effects solely to ALA treatment, raising concerns about the validity of the findings.
In conclusion, while the study presents interesting results regarding ALA's effect on pain intensity and oxidative stress markers in diabetic patients, its major limitations—small sample size, short treatment duration, and absence of a placebo group—undermine the strength of its conclusions. Addressing these issues in future research will be crucial for determining ALA's true therapeutic potential.
Comments on the Quality of English Languagemoderat
Author Response
We thank Reviewer 1 for valuable comments and suggestions.
Comment 1: Small Sample Size: The study involved only 16 diabetic patients, which is a small sample size to draw strong, generalizable conclusions. This limitation is also mentioned regarding the analysis of biomarkers like 8-OHdG (only nine subjects analyzed post-treatment).
Response: We added new paragraph to our manuscript (Limitations) where we are discussing and explaining these limitations and their influence to the interpretation of our data.
Comment 2: Limited Duration of Treatment: The study used a short-term treatment duration (4-9 days), which may not be sufficient to observe long-term effects of α-lipoic acid (ALA) on DNA damage markers like SCEs and MN. This could explain the lack of significant changes in these markers.
Response: Our study was designed not as clinical trial but as a pilot observational study dealing with real-world situations. In real-world situation neither physicians nor patients have possibility to receive intravenous ALA treatment for weeks – usual practice is to deliver this drug for a few days or, in the best case, a few weeks. Additional explanations are now included into the text describing goals and design of our study more clearly.
Comment 3: No Significant Changes in Cytogenetic Markers (MN and SCE): Despite the reduction in pain intensity and 8-OHdG levels, the study did not observe significant changes in MN and SCE frequencies. This is crucial as it raises concerns about whether ALA is truly effective at reducing DNA damage over such a short treatment window.
Response: We agree that the question about effectivity of ALA in reducing such secondary DNA damage as MN remains open. Sentences discussing this is included into manuscript.
Comment 4: Lack of Control Group for ALA Treatment: There was no control group to compare the effect of ALA treatment in diabetic patients. Including a placebo group could help clarify the specific impact of ALA compared to other treatments or no treatment.
Response: Again, our study was designed not as clinical trial but as a pilot observational study dealing with real-world situations. Since it is not a registered clinical trial, we have no legal possibility to include control or placebo groups in our study. We agree that such groups will be of crucial importance in further studies. Additional explanations are now included into the text describing goals and design of our study more clearly. Some aspects are also discussed in newly added Limitations paragraph.
Comment 5: Discussion of Conflicting Literature: The authors mention contradictory results from meta-analyses regarding ALA's effectiveness in reducing neuropathic pain. This conflict should be discussed more comprehensively to explain how the present study fits into the broader literature.
Response: Additional explanations are now included into the text
Comment 6: Figure 1: The representation of pain intensity in subjects before and after treatment is clear, but it is suggested to use error bars or standard deviations to improve visual interpretation of variability across the population.
Response: Figure 1 shows all individual values of pain intensity before and after treatment, thus variability across the population could be seen from these individual values. On our opinion, adding box-plots, mean values for the group or SDs will make interpretation of the results more complicated.
Comment 7: Figure 3: The microphotographs included in the figure (e.g., SCEs and MN) need higher resolution or improved contrast to make them easier to interpret. In their current form, they are difficult to analyze, especially for readers unfamiliar with these techniques.
Response: High-resolution images are uploaded to the manuscript submission system. We can’t use these pictures directly in manuscript because the file will be too large.
Comment 8: Methodological Detail: The study briefly mentions the biomarkers and techniques (e.g., ELISA, microscopy), but more detailed protocols should be provided for reproducibility, particularly regarding the quantification of biomarkers such as SCEs and MN.
Response: We added additional details to the protocols of SCEs and MN.
Comment 9: Inconsistency in Results Reporting: The results section does not consistently report the demographic and clinical characteristics of the study subjects, especially considering the gender and age of the participants. Demographic information is critical when interpreting pain perception and treatment effects.
Response: Demographic and clinical characteristics of the study subjects are shown in Supplementary Tables 1 and 2. However, in the revised version we also included some explanatory sentences in the Results section. Besides, some additional indices (body mass index and number of comorbid conditions were included in Supplementary Tables.
Comment 10: Statistical Power and Sample Size: Given the small sample size, the study lacks sufficient statistical power to detect minor differences, especially in DNA damage markers. This is a serious limitation that affects the reliability of the conclusions drawn, particularly in terms of generalizing the findings.
Response: We agree that small sample size is a problem with SCE and MN data and is a probable reason why we have not seen any significant differences after ALA treatment. We added note regarding this issue in new section Limitations. However, in case of pairwise comparisons of pain and 8-OHdG before and after ALA treatment, we observed very large effect size – 0.88 and 0.96, respectively. We believe that such a large effect size supports reliability of our findings despite of a small sample size. We added data about effect size into the text.
Comments 11 and 12: Lack of Long-Term Follow-up: The short duration of the study (4-9 days) is insufficient for determining the long-term effectiveness of ALA in treating diabetic neuropathy and its impact on biomarkers of DNA damage. Future studies should consider a longer treatment and follow-up period to capture the full extent of ALA’s effects.
Absence of Placebo-Controlled Design: A placebo-controlled design is essential in clinical trials to eliminate bias. The lack of such a group makes it difficult to attribute observed effects solely to ALA treatment, raising concerns about the validity of the findings.
Response: Our study was designed not as clinical trial but as a pilot observational study dealing with real-world situations. In real-world situation neither physicians nor patients have possibility to receive intravenous ALA treatment for weeks – usual practice is to deliver this drug for a few days or, in the best case, a few weeks. Since our study is not a registered clinical trial, we have no legal possibility to include control or placebo groups in our study. We agree that such groups will be of crucial importance in further studies. Additional explanations are now included into the text describing goals and design of our study more clearly. Some aspects are also discussed in newly added Limitations section.
Reviewer 2 Report
Comments and Suggestions for Authors
Multidisciplinary Digital Publishing Institute (MDPI): pharmaceuticals-3254910
Title: Articles
Effects of Short-Term Treatment with a-Lipoic Acid on Neuropathic Pain and Biomarkers of DNA Damage in Patients with Diabetes Mellitus
Overall comments:
The authors decided to assess the efficacy of the use of short-term intravenous (IV) administration of alpha lipoic acid (ALA) to for neuropathic pain in individuals with diabetes mellitus (DM), type I and type II DM, in which oxidative stress has been noted to have the main causal role. Therefore, there have been a number of clinical studies on ALA, but in those, a long-term administration has been assessed.
The authors decided to assess the efficacy of short-term IV administration of ALA via several parameters such as subjective pain perception using the Universal Pain Assessment Tool (UPAT) as well as objective measures of biomarkers of DNA damage to determine if any changes occur in the concentration of 8-hydroxy-2-guanosine (8-OHdG), as well as micronuclei (MN) and sister-chromatic exchanges (SCEs) in lymphocytes.
Recommend not to mix type I (T1) and type II (T2) DM because T1DM is although it results in metabolic derangements, but it is an autoimmune disease. Therefore, the mixing them would muddle any results one obtains in my opinion. There are many aspects of two (although more than 2 in actuality), are different. Although once the metabolic derangements occur, complications may be similar, but it may be possible that pathophysiology may not be the same.
Alternatively, it may be good to have two sub-groups to determine the differences in responses to ALA. Need more patients with T1DM.
Treatments, especially in the early phases are different in two, and diabetic ketoacidosis is a big risk factor for T1DM, not usually in T2DM.
A flow chart of the study would be helpful in understand how the study was performed.
Please also discuss what is the underlying mechanisms that ALA is effective in ameliorating neuropathy?
Many researchers state that ALA works in neuropathy by reducing oxidative stress (OS), improving nerve blood flow, and increasing the levels of antioxidants. However, it misses what is the reason for increased reactive oxygen species as well as how specifically ALA works to ameliorate OS.
However, even when conducting clinical studies, it is important to have some insight into the underlying biology beyond what are accessible through metabolites or cells which can be observed.
Melli G, Taiana M, Camozzi F, Triolo D, Podini P, Quattrini A, Taroni F, Lauria G. Alpha-lipoic acid prevents mitochondrial damage and neurotoxicity in experimental chemotherapy neuropathy. Exp Neurol. 2008 Dec;214(2):276-84. doi: 10.1016/j.expneurol.2008.08.013. Epub 2008 Sep 9. PMID: 18809400.
Longhitano L, Distefano A, Amorini AM, Orlando L, Giallongo S, Tibullo D, Lazzarino G, Nicolosi A, Alanazi AM, Saoca C, Macaione V, Aguennouz M, Salomone F, Tropea E, Barbagallo IA, Volti GL, Lazzarino G. (+)-Lipoic Acid Reduces Lipotoxicity and Regulates Mitochondrial Homeostasis and Energy Balance in an In Vitro Model of Liver Steatosis. Int J Mol Sci. 2023 Sep 23;24(19):14491. doi: 10.3390/ijms241914491. PMID: 37833939; PMCID: PMC10572323.
Dos Santos SM, Romeiro CFR, Rodrigues CA, Cerqueira ARL, Monteiro MC. Mitochondrial Dysfunction and Alpha-Lipoic Acid: Beneficial or Harmful in Alzheimer's Disease? Oxid Med Cell Longev. 2019 Nov 30;2019:8409329. doi: 10.1155/2019/8409329. PMID: 31885820; PMCID: PMC6914903.
Specific comments:
Abstract:
Lines 14-16: Recommend adding “insulin resistance” in the description of diabetes mellitus (DM) although because of type 1 mixing, this probably was taken out.
It may be a bit incomplete to just list macro- and micro-vascular complications since there are many others.
Recommend,
Various complications, some of which are macro- and microvascular complications, so that the readers do not assume that they are the only complications.
The most common microvascular complication is diabetic retinopathy, not neuropathy.
1.
Introduction:
Lines 34-35: Please see the suggestions for the abstract. Revising in the main text would be particularly important.
Line 38: Suggest briefly describing the symptoms often associated with diabetic peripheral neuropathy (DPN) or diabetic polyneuropathy?
Bodman MA, Dreyer MA, Varacallo M. Diabetic Peripheral Neuropathy. [Updated 2024 Feb 25]. In: StatPearls [Internet]. Treasure Island (FL): StatPearls Publishing; 2024 Jan-. Available from: https://www.ncbi.nlm.nih.gov/books/NBK442009/
Peripheral neuropathy often presents with varying degrees of numbness, tingling, aching, burning sensation, weakness of limbs, hyperalgesia, allodynia, and pain. Neuropathic pain has been characterized as superficial, deep-seated, or severe, unremitting pain with exacerbation at night.
Lines 39-41: Suggest replacing the words, “the painful variant” to another words, such as neuropathic pain or the associated pain etc.
Suggest revising, “…are linked to sleep..”
“… have a profound impact on quality of life, resulting in sleep disturbances and difficulties in the activities of daily living, as well as increasing morbidity and mortality in individuals with DM.”
Lines 42-43: Would replace “A lot of to “numerous” or simply “many”.
Numerous studies have reported that …
Please delete “(“ or add “)” as needed.
Lines 48-51: Suggest replacing, “Besides” to “In addition”.
“In addition, other studies have reported that …”
Please describe briefly about oxidative stress.
What are the causes of oxidative stress?
Oxidative stress is typically brought on by inciting events or factors.
Pizzino G, Irrera N, Cucinotta M, Pallio G, Mannino F, Arcoraci V, Squadrito F, Altavilla D, Bitto A. Oxidative Stress: Harms and Benefits for Human Health. Oxid Med Cell Longev. 2017;2017:8416763. doi: 10.1155/2017/8416763. Epub 2017 Jul 27. PMID: 28819546; PMCID: PMC5551541.
Maynard S, Schurman SH, Harboe C, de Souza-Pinto NC, Bohr VA. Base excision repair of oxidative DNA damage and association with cancer and aging. Carcinogenesis. 2009 Jan;30(1):2-10. doi: 10.1093/carcin/bgn250. Epub 2008 Oct 31. PMID: 18978338; PMCID: PMC2639036.
8-OHdG is a product of oxidative damage to 2′-deoxyguanosine, a DNA nucleobase. It's a major form of oxidative DNA damage caused by free radicals in nuclear and mitochondrial DNA.
Where do reactive oxygen species adioactive oxygenROS
Snezhkina AV, Kudryavtseva AV, Kardymon OL, Savvateeva MV, Melnikova NV, Krasnov GS, Dmitriev AA. ROS Generation and Antioxidant Defense Systems in Normal and Malignant Cells. Oxid Med Cell Longev. 2019 Aug 5;2019:6175804. doi: 10.1155/2019/6175804. PMID: 31467634; PMCID: PMC6701375.
Lines 52-55:
It is also important to know that ROS are important signaling molecules; however, in excess they are harmful.
The authors should mention about ROS.
Please briefly describe known antioxidant molecules.
Such as
Antioxidant molecules that can help reduce oxidative stress include:
•
Enzymatic antioxidants: superoxide dismutase (SOD), catalase (CAT), and glutathione peroxidase (GPx) work in the mitochondria.
•
Non-enzymatic antioxidants: vitamin C, vitamin E, flavonoids, carotenoids, melatonin, ergothioneine, lipoic acid, glutathione, coenzyme Q10 (CoQ10) can scavenge excess ROS.
•
Exogenous antioxidants: diets or supplements.
•
Resveratrol and sulforaphane.
Oxidative stress results from an imbalance between the generation and amelioration of ROS which can lead to cell damages and ROS signaling disruption.
Please describe briefly about alpha-lipoic acid in the introduction beyond what are listed in Lines 59-63.
What the authors listed in Lines 57-59 are not the direct effects of alpha lipoic acid, and they are secondary positive consequences. They should be distinguished and should not be confused with the direct effects of alpha lipoic acid.
Alpha-Lipoic Acid
Nguyen H, Pellegrini MV, Gupta V. Alpha-Lipoic Acid. [Updated 2024 Jan 26]. In: StatPearls [Internet]. Treasure Island (FL): StatPearls Publishing; 2024 Jan-. Available from: https://www.ncbi.nlm.nih.gov/books/NBK564301/.
•
Alpha-lipoic acid (ALA) is a caprylic acid-derived antioxidant.
•
This organosulfur compound is synthesized in the mitochondria and is a cofactor in the enzymatic nutrient breakdown.
•
ALA is also available in red meat, beets, carrots, potatoes, spinach, and broccoli. ALA consists of a dithiol functional group that eliminates reactive oxygen species (ROS) by reducing the oxidized forms of other antioxidants.
Han T, Bai J, Liu W, et al. A systematic review and meta-analysis of alpha-lipoic acid in the treatment of diabetic peripheral neuropathy. 2012. In: Database of Abstracts of Reviews of Effects (DARE): Quality-assessed Reviews [Internet]. York (UK): Centre for
Reviews and Dissemination (UK); 1995-. Available from: https://www.ncbi.nlm.nih.gov/books/NBK114396/.
Suggest reading the article below to understand the positive effects of alpha-lipoic acid.
Melli G, Taiana M, Camozzi F, Triolo D, Podini P, Quattrini A, Taroni F, Lauria G. Alpha-lipoic acid prevents mitochondrial damage and neurotoxicity in experimental chemotherapy neuropathy. Exp Neurol. 2008 Dec;214(2):276-84. doi: 10.1016/j.expneurol.2008.08.013. Epub 2008 Sep 9. PMID: 18809400.
Although sister chromatid exchange abnormalities can be seen with oxidative stress, it was not clear why this and micronuclei were selected among other biomarkers of oxidative stress. ROS associated biomarkers would have been more suitable?
Frijhoff J, Winyard PG, Zarkovic N, Davies SS, Stocker R, Cheng D, Knight AR, Taylor EL, Oettrich J, Ruskovska T, Gasparovic AC, Cuadrado A, Weber D, Poulsen HE, Grune T, Schmidt HH, Ghezzi P. Clinical Relevance of Biomarkers of Oxidative Stress. Antioxid Redox Signal. 2015 Nov 10;23(14):1144-70. doi: 10.1089/ars.2015.6317. Epub 2015 Oct 26. PMID: 26415143; PMCID: PMC4657513.
2.
Results:
3.
Discussion:
It is important to choose individuals in the control group carefully because oxidative stress or high ROS can be associated with many other disease processes. It would have been important to measure another type of inflammatory marker in patients which is easily accessible in patients, such as high-sensitive C-reactive protein (hsCRP) or erythrocyte sedimentation rate (ESR).
No demographic information on the participants was provided so that if any of them has a history of cancer or any other chronic diseases in addition. It was not clear from the texts.
Furthermore, having both type I and type II diabetes mellitus (DM) is not a good idea since the underlying pathophysiology of DM is different in two types. Type I DM is a type of autoimmune disorder.
It was interesting that “Materials and Methods” section was after the discussion and conclusion sections. I am not sure this is the organization this particular journal utilizes, but it is more helpful to the readers to have them prior to the study results.
Thank you very much for allowing me to review this manuscript.
Sincerely,
Comments on the Quality of English LanguageOverall, written English was fine and minor editing would be helpful.
Author Response
We are sincerely thank Reviewer 2 for valuable comments and suggestions that certainly will improve our work.
Comment 1: Recommend not to mix type I (T1) and type II (T2) DM because T1DM is although it results in metabolic derangements, but it is an autoimmune disease. Therefore, the mixing them would muddle any results one obtains in my opinion. There are many aspects of two (although more than 2 in actuality), are different. Although once the metabolic derangements occur, complications may be similar, but it may be possible that pathophysiology may not be the same. Alternatively, it may be good to have two sub-groups to determine the differences in responses to ALA. Need more patients with T1DM.
Response: Although there are some epidemiological differences between diabetic polyneuropathy in T1DM and T2DM patients, nerve pathology, mechanisms of their damage, and main symptoms are the same (see, for example, Pop-Busui, R.; Ang, L.; Boulton, A.; Feldman, E.; Marcus, R.; Mizokami-Stout, K.; Singleton, J.R.; Ziegler, D. Diagnosis and Treatment of Painful Diabetic Peripheral Neuropathy. ADA Clinical Compendia 2022, 2022, 1–32, doi:10.2337/db2022-01). We added corresponding note into the text of our manuscript.
Comment 2: Treatments, especially in the early phases are different in two, and diabetic ketoacidosis is a big risk factor for T1DM, not usually in T2DM.
Response: Average disease duration was 15.7 years for T2DM patients and 13.7 years for T1DM, and none of the subjects were at the early phases of disease. Corresponding explanations are added into the text.
Comment 3: A flow chart of the study would be helpful in understand how the study was performed.
Response: Study design was quite simple – two blood sampling points and measuring the same indices at these points. We feel that making such simple flow-chart will be in excess.
Comment 4: Please also discuss what is the underlying mechanisms that ALA is effective in ameliorating neuropathy? Many researchers state that ALA works in neuropathy by reducing oxidative stress (OS), improving nerve blood flow, and increasing the levels of antioxidants. However, it misses what is the reason for increased reactive oxygen species as well as how specifically ALA works to ameliorate OS. However, even when conducting clinical studies, it is important to have some insight into the underlying biology beyond what are accessible through metabolites or cells which can be observed
Response: Thank you for good recommendation, short discussion about underlying mechanisms of ALA action is added.
Comment 5: Lines 14-16: Recommend adding “insulin resistance” in the description of diabetes mellitus (DM) although because of type 1 mixing, this probably was taken out.
Response: Corrected
Comment 6: It may be a bit incomplete to just list macro- and micro-vascular complications since there are many others. Recommend, Various complications, some of which are macro- and microvascular complications, so that the readers do not assume that they are the only complications
Response: Corrected
Comment 7: The most common microvascular complication is diabetic retinopathy, not neuropathy
Response:
This issue is quite complicated because there are contradictory data about what complication is the most common. Thus, we changed our statement to: “Diabetic polyneuropathy (DPN) is the most common microvascular diabetic complication...“
Comment 8: Lines 34-35: Please see the suggestions for the abstract. Revising in the main text would be particularly important
Response: Corrected
Comment 9: Line 38: Suggest briefly describing the symptoms often associated with diabetic peripheral neuropathy (DPN) or diabetic polyneuropathy?
Response: Added
Comment 10: Lines 39-41: Suggest replacing the words, “the painful variant” to another words, such as neuropathic pain or the associated pain etc.
Response: Corrected
Comment 11: Suggest revising, “…are linked to sleep..” “… have a profound impact on quality of life, resulting in sleep disturbances and difficulties in the activities of daily living, as well as increasing morbidity and mortality in individuals with DM.”
Response: Corrected
Comment 12: Lines 42-43: Would replace “A lot of to “numerous” or simply “many”. Numerous studies have reported that … Please delete “(“ or add “)” as needed.
Response: Corrected
Comment 13: Lines 48-51: Suggest replacing, “Besides” to “In addition”. “In addition, other studies have reported that …”
Response: Corrected
Comment 14: Please describe briefly about oxidative stress. What are the causes of oxidative stress? Oxidative stress is typically brought on by inciting events or factors.
Response: Short description added
Comment 15: Lines 52-55: It is also important to know that ROS are important signaling molecules; however, in excess they are harmful. The authors should mention about ROS.
Response: Corrected
Comment 16: Please briefly describe known antioxidant molecules. Such as Antioxidant molecules that can help reduce oxidative stress include: Enzymatic antioxidants: superoxide dismutase (SOD), catalase (CAT), and glutathione peroxidase (GPx) work in the mitochondria. Non-enzymatic antioxidants: vitamin C, vitamin E, flavonoids, carotenoids, melatonin, ergothioneine, lipoic acid, glutathione, coenzyme Q10 (CoQ10) can scavenge excess ROS. Exogenous antioxidants: diets or supplements. Resveratrol and sulforaphane. Oxidative stress results from an imbalance between the generation and amelioration of ROS which can lead to cell damages and ROS signaling disruption.
Response: Short description added
Comment 17: Please describe briefly about alpha-lipoic acid in the introduction beyond what are listed in Lines 59-63. What the authors listed in Lines 57-59 are not the direct effects of alpha lipoic acid, and they are secondary positive consequences. They should be distinguished and should not be confused with the direct effects of alpha lipoic acid.
Response: Brief description added
Comment 18: Although sister chromatid exchange abnormalities can be seen with oxidative stress, it was not clear why this and micronuclei were selected among other biomarkers of oxidative stress. ROS associated biomarkers would have been more suitable?
Response: As an immediate biomarker of oxidative stress, we used 8-OHdG. SCEs and MN were selected as biomarkers of later effects of oxidative stress: SCEs are good indicator of genomic instability, and increase frequency of MN could be related to higher cancer risk (which is well known for diabetic subjects). Corresponding explanations and references are added to the text.
Comment 19: It is important to choose individuals in the control group carefully because oxidative stress or high ROS can be associated with many other disease processes. It would have been important to measure another type of inflammatory marker in patients which is easily accessible in patients, such as high-sensitive C-reactive protein (hsCRP) or erythrocyte sedimentation rate (ESR).
Response: Although large meta-analysis on the effects of ALA on CRP has shown that ALA is able to reduce CRP levels only when treatment duration is longer than 8 weeks and only in non-diabetic patients (Saboori, S.; Falahi, E.; Eslampour, E.; Zeinali Khosroshahi, M.; Yousefi Rad, E. Effects of Alpha-Lipoic Acid Supplementation on C-Reactive Protein Level: A Systematic Review and Meta-Analysis of Randomized Controlled Clinical Trials. Nutrition, Metabolism and Cardiovascular Diseases 2018, 28, 779–786, doi:10.1016/j.numecd.2018.04.003.), it is really good idea and we will include these markers in our future studies.
Comment 20: No demographic information on the participants was provided so that if any of them has a history of cancer or any other chronic diseases in addition. It was not clear from the texts.
Response: Demographic characteristics of the participants were shown in Supplementary tables 1 and 2. However, now we also included some explanations in the main text. We also added some additional characteristics (body mass index and number of comorbid conditions) to both supplementary tables.
Comment 21: Furthermore, having both type I and type II diabetes mellitus (DM) is not a good idea since the underlying pathophysiology of DM is different in two types. Type I DM is a type of autoimmune disorder
Response: Please, see response to the reviewer’s comment #1
Comment 22: It was interesting that “Materials and Methods” section was after the discussion and conclusion sections. I am not sure this is the organization this particular journal utilizes, but it is more helpful to the readers to have them prior to the study results.
Response: “Materials and Methods” section were placed after the discussion according to the journal’s style
Round 2
Reviewer 1 Report
Comments and Suggestions for Authors
Thank for your response
My comments may seem rough, but the journal is Q1 and the level of the studies should be highly judged and revised.
Thank you again
Congratulation
Comments on the Quality of English Languagegood
Author Response
Comment: My comments may seem rough, but the journal is Q1 and the level of the studies should be highly judged and revised
Response: We wish to thank Reviewer 1 for efforts and time spent while reviewing our manuscript. We completely agree about the requirements that should be applied for clinical trials. However, as we explained in our previous response, our study is not a clinical trial, it is a small real-world study that corresponds to the realities of clinical practice. Consequently, we were trying to answer the question "Whether short-term ALA treatment could be beneficial for the patient"? rather than the question "Is ALA effective drug for the management of diabetic polyneuropathy"? We tried to do our best improving the manuscript, however, we are unable to completely redesign the whole study and make it a clinical trial. We still hope that our study, despite of its small sample size, will be interesting for the readers of Pharmaceuticals.
Reviewer 2 Report
Comments and Suggestions for Authors
Multidisciplinary Digital Publishing Institute (MDPI): pharmaceuticals-3254910
Title: Articles
Effects of Short-Term Treatment with a-Lipoic Acid on Neuropathic Pain and Biomarkers of DNA Damage in Patients with Diabetes Mellitus
Overall comments:
The authors decided to assess the efficacy of the use of short-term intravenous (IV) administration of alpha lipoic acid (ALA) to for neuropathic pain in individuals with diabetes mellitus (DM), type I and type II DM, in which oxidative stress has been noted to have the main causal role. Therefore, there have been a number of clinical studies on ALA, but in those, a long-term administration has been assessed.
The authors decided to assess the efficacy of short-term IV administration of ALA via several parameters such as subjective pain perception using the Universal Pain Assessment Tool (UPAT) as well as objective measures of biomarkers of DNA damage to determine if any changes occur in the concentration of 8-hydroxy-2-guanosine (8-OHdG), as well as micronuclei (MN) and sister-chromatic exchanges (SCEs) in lymphocytes.
Strongly recommending not to mix type I (T1) and type II (T2) DM because T1DM is although it results in metabolic derangements, it is an autoimmune disease.
Therefore, mixing them would muddle any results one obtains in my opinion.
There are many aspects of two (although there are more than 2 in actuality), are distinctively different.
Although once the metabolic derangements occur, complications seem to be similar, its pathophysiology may not be the same.
Please review the article below:
Sempere-Bigorra M, Julián-Rochina I, Cauli O. Differences and Similarities in Neuropathy in Type 1 and 2 Diabetes: A Systematic Review. J Pers Med. 2021 Mar 22;11(3):230. doi: 10.3390/jpm11030230. PMID: 33810048; PMCID: PMC8004786.
Sima AA, Kamiya H. Diabetic neuropathy differs in type 1 and type 2 diabetes. Ann N Y Acad Sci. 2006 Nov;1084:235-49. doi: 10.1196/annals.1372.004. PMID: 17151305.
Feldman, E.L., Callaghan, B.C., Pop-Busui, R. et al. Diabetic neuropathy. Nat Rev Dis Primers 5, 41 (2019). https://doi.org/10.1038/s41572-019-0092-1.
Alternatively, it may be good to have two sub-groups to determine the differences in responses to ALA. Need more patients with T1DM.
If you have ever treated DM, both type I and type II, one will know that they are totally different in many aspects, and they cannot be mixed together even when discussing complications. How patients present with complications is totally different.
It is not appropriate to mix two types of DM in any study from the genetics point of view either. Therefore, they have to be studied separately.
Thank you very much for allowing me to review this manuscript.
Sincerely,
Comments on the Quality of English LanguageSome editing would improve its readability.
Author Response
Comment 1: Strongly recommending not to mix type I (T1) and type II (T2) DM because T1DM is although it results in metabolic derangements, it is an autoimmune disease. Therefore, mixing them would muddle any results one obtains in my opinion. There are many aspects of two (although there are more than 2 in actuality), are distinctively different. Although once the metabolic derangements occur, complications seem to be similar, its pathophysiology may not be the same. Alternatively, it may be good to have two sub-groups to determine the differences in responses to ALA. Need more patients with T1DM.
Response: We wish to thank Reviewer 2 for very useful comments and suggestions. Of course, we agree that T1DM and T2DM are different diseases. In our team we have experienced medical doctors that have treated T1DM and T2DM patients for decades. Our initial intention was to consider patients of both types of diabetes as one group mainly for some reasons: 1) target group was small; 2) alpha-lipoic acid is recommended for DPN management in both T1DM and T2DM patients; 3) ALA is used as a disease modifying therapy targeting pathogenesis of DPN related to oxidative stress and mitochondrial damage. However, we agree that results obtained in subgroups of T1DM and T2DM should be discussed as well. So, we modified Figures 1, 2 and 3 and the whole Results section accordingly. Some additional explanations was added to the text as well.
Round 3
Reviewer 2 Report
Comments and Suggestions for Authors
Multidisciplinary Digital Publishing Institute (MDPI): pharmaceuticals-3254910_v3
Title: Articles
Effects of Short-Term Treatment with a-Lipoic Acid on Neuropathic Pain and Biomarkers of DNA Damage in Patients with Diabetes Mellitus
Overall comments:
The authors decided to assess the efficacy of the use of short-term intravenous (IV) administration of alpha lipoic acid (ALA) to for neuropathic pain in individuals with diabetes mellitus (DM), type I and type II DM, in which oxidative stress has been noted to have the main causal role. Therefore, there have been a number of clinical studies on ALA, but in those, long-term administration has been assessed.
The authors decided to assess the efficacy of short-term IV administration of ALA via several parameters such as subjective pain perception using the Universal Pain Assessment Tool (UPAT) as well as objective measures of biomarkers of DNA damage to determine if any changes occur in the concentration of 8-hydroxy-2-guanosine (8-OHdG), as well as micronuclei (MN) and sister-chromatic exchanges (SCEs) in lymphocytes.
Although the authors have tried to improve the manuscript by trying to separate type I and type II diabetes mellitus (T1DM) and (T2DM), respectively. From the manuscript, the revised manuscript still does not convey the message that there is much difference between T1DM and T2DM.
It is important to show that the authors describe the differences in the introduction.
It seems as though not much has been added to describe the differences between two types, in regard to diabetic polyneuropathy.
It is also important to describe that T1DM is an “autoimmune disease”. Thus, the abstract requires editing.
Insulin resistance is typically only observed in T2DM, so it is incorrect to say that:
Lines 35-36: Diabetes mellitus (DM) is a metabolic disorder characterized by persistent hyperglycemia and insulin resistance in type II DM, not type I DM.
(please revise this in the abstract also).
Sempere-Bigorra M, Julián-Rochina I, Cauli O. Differences and Similarities in Neuropathy in Type 1 and 2 Diabetes: A Systematic Review. J Pers Med. 2021 Mar 22;11(3):230. doi: 10.3390/jpm11030230. PMID: 33810048; PMCID: PMC8004786.
André Pfannkuche, Ahmad Alhajjar, Antao Ming, Isabell Walter, Claudia Piehler, Peter R. Mertens, Prevalence and risk factors of diabetic peripheral neuropathy in a diabetics cohort: Register initiative “diabetes and nerves”, Endocrine and Metabolic Science, Volume 1, Issues 1–2, 2020, 100053, ISSN 2666-3961, https://doi.org/10.1016/j.endmts.2020.100053. (https://www.sciencedirect.com/science/article/pii/S2666396120300078)
The overall diabetic peripheral neuropathy prevalence of 40.3% in patients with T2DM, and 29.1% in T1DM.
Furthermore, some report that more patients present with DPN earlier in the diagnosis of T2DM than T1DM. In T1DM, it may be years before patients present with DPN.
Line 47: Recommend replacing “production” with another word such as “generation”.
Even though in the normal conditions, reactive oxygen species may play important roles as the authors describe, “generation” is more appropriate than “production”.
Thanan R, Oikawa S, Hiraku Y, Ohnishi S, Ma N, Pinlaor S, Yongvanit P, Kawanishi S, Murata M. Oxidative stress and its significant roles in neurodegenerative diseases and cancer. Int J Mol Sci. 2014 Dec 24;16(1):193-217. doi: 10.3390/ijms16010193. PMID: 25547488; PMCID: PMC4307243.
Line 55: Replacing “the multifaceted role(s)” to “the multifaceted effects in the pathogenesis of DPN, primarily impacting nerve cell integrity.”
Line 56: Recommend revising, “nerve cells structures”.
Either “nerve cell structures” or “nerve cells”
Recommend replacing “, including” to “, in addition to alterations to lipids, proteins and DNA.”
Recommend elaborating more on anti-oxidative mechanisms since they are important in conjunction with the function of ALA.
Furthermore, it may also be informative to the readers to explain why ROS may be more abundant in T2DM, as well as late stages of T1DM.
Checa J, Aran JM. Reactive Oxygen Species: Drivers of Physiological and Pathological Processes. J Inflamm Res. 2020 Dec 2;13:1057-1073. doi: 10.2147/JIR.S275595. PMID: 33293849; PMCID: PMC7719303.
Halliwell B. Understanding mechanisms of antioxidant action in health and disease. Nat Rev Mol Cell Biol. 2024 Jan;25(1):13-33. doi: 10.1038/s41580-023-00645-4. Epub 2023 Sep 15. PMID: 37714962.
Lines 64-94: This paragraph is critically important in convincing the readers why ALA would be more efficacious in DPN than other anti-oxidants or other mechanisms. Beyond what the clinical trials have shown in the past, it may be important to discuss this in depth.
Lines 75-80: This section is critical in explaining why ALA may be a good molecule to ameliorate insulin insensitivity or other complications. Even though the authors tried to connect ROS excess to DPN, it is more desirable if more biological mechanisms are well explained to the readers.
Murphy MP. How mitochondria produce reactive oxygen species. Biochem J. 2009 Jan 1;417(1):1-13. doi: 10.1042/BJ20081386. PMID: 19061483; PMCID: PMC2605959.
Dos Santos SM, Romeiro CFR, Rodrigues CA, Cerqueira ARL, Monteiro MC. Mitochondrial Dysfunction and Alpha-Lipoic Acid: Beneficial or Harmful in Alzheimer's Disease? Oxid Med Cell Longev. 2019 Nov 30;2019:8409329. doi: 10.1155/2019/8409329. PMID: 31885820; PMCID: PMC6914903.
Packer L, Witt EH, Tritschler HJ. alpha-Lipoic acid as a biological antioxidant. Free Radic Biol Med. 1995 Aug;19(2):227-50. doi: 10.1016/0891-5849(95)00017-r. PMID: 7649494.
The authors should describe how DM, especially T2DM connects to ROS excess, insulin resistance, and positive effects of ALA as antioxidant molecules.
From the paragraph written, the connection is not discussed well in biological terms.
Lines 85-96: It would be important to provide more details on pharmacokinetics and pharmacology of ALA (Lines 87-89) so that the authors’ argument for the use of ALA for less than 10 days could be a practical option (the current argument is too vague).
Overall, agree with the use of ALA in the attempt to ameliorate the symptoms associated with DPN, but it is important to have the introduction written with the underlying biological processes being considered.
Yu Y, Xu J, Li H, Lv J, Zhang Y, Niu R, Wang J, Zhao Y, Sun Z. α-Lipoic acid improves mitochondrial biogenesis and dynamics by enhancing antioxidant and inhibiting Wnt/Ca2+ pathway to relieve fluoride-induced hepatotoxic injury. Chem Biol Interact. 2023 Nov 1;385:110719. doi: 10.1016/j.cbi.2023.110719. Epub 2023 Sep 20. PMID: 37739047.
Furthermore, the authors still did not make the distinction between T1DM and T2DM well in the introduction, especially not mentioning T1DM is an autoimmune disease. It is not typically associated with insulin resistance.
3. Discussion:
Line 206: Suggest starting a new paragraph at, “It is interesting to note …”
At this point, did the authors decide to examine the same pain parameters or any change in biomarkers studied?
It may be important to note these so that it may be possible to determine the most appropriate length of treatment without reducing the effectiveness of ALA therapy.
In the discussion, the authors described that It is also important to note how long the pain reduction lasted. This information would be critically important if this administration were to be clinically considered, one needs to know how long the effectiveness lasts.
Thank you very much for allowing me to review this manuscript.
Sincerely,

Some suggestions have been made in the report above.
Some editing would improve the quality of the manuscript.
Author Response
We wish to thank Reviewer 2 for valuable comments and suggestions that certainly helped us to improve our manuscript.
Comment 1: It is important to show that the authors describe the differences in the introduction. It seems as though not much has been added to describe the differences between two types, in regard to diabetic polyneuropathy. It is also important to describe that T1DM is an “autoimmune disease”. Thus, the abstract requires editing. Insulin resistance is typically only observed in T2DM, so it is incorrect to say that: Lines 35-36: Diabetes mellitus (DM) is a metabolic disorder characterized by persistent hyperglycemia and insulin resistance in type II DM, not type I DM. (please revise this in the abstract also).
Response: All these issues are corrected both in abstract and introduction
Comment 2: The overall diabetic peripheral neuropathy prevalence of 40.3% in patients with T2DM, and 29.1% in T1DM. Furthermore, some report that more patients present with DPN earlier in the diagnosis of T2DM than T1DM. In T1DM, it may be years before patients present with DPN.
Response: In introduction, we added information about the prevalence of DPN in T2DM and T1DM, However, we used numbers not from the single study (recommended by Reviewer) but from the paper where of several studies are reviewed (Ref.2)
Comment 3: Line 47: Recommend replacing “production” with another word such as “generation”. Even though in the normal conditions, reactive oxygen species may play important roles as the authors describe, “generation” is more appropriate than “production”.
Response: Corrected
Comment 4: Line 55: Replacing “the multifaceted role(s)” to “the multifaceted effects in the pathogenesis of DPN, primarily impacting nerve cell integrity.” Line 56: Recommend revising, “nerve cells structures”. Either “nerve cell structures” or “nerve cells”. Recommend replacing “, including” to “, in addition to alterations to lipids, proteins and DNA.”
Response: Corrected
Comment 5: Recommend elaborating more on anti-oxidative mechanisms since they are important in conjunction with the function of ALA.
Response: We agree that knowing anti-oxidative mechanisms of ALA and other antioxidants' action is very important. However, since in our paper we have not investigated anti-oxidative mechanisms of ALA action, we feel that deep analysis of these mechanisms in introduction will be somewhat excessive. Instead, we put some references to review articles where these mechanisms are discussed in depth.
Comment 6: Furthermore, it may also be informative to the readers to explain why ROS may be more abundant in T2DM, as well as late stages of T1DM.
Response: In revised version of the manuscript we discussed main differences between T1DM and T2DM in relation to DPN. Please note, however, that elucidation of differences in T1DM and T2DM was not the goal of our research, moreover, our data do not indicate significant differences in response to ALA treatment between T1DM and T2DM patients.
Comment 7: Lines 64-94: This paragraph is critically important in convincing the readers why ALA would be more efficacious in DPN than other anti-oxidants or other mechanisms. Beyond what the clinical trials have shown in the past, it may be important to discuss this in depth. Lines 75-80: This section is critical in explaining why ALA may be a good molecule to ameliorate insulin insensitivity or other complications. Even though the authors tried to connect ROS excess to DPN, it is more desirable if more biological mechanisms are well explained to the readers.
Response: In order to keep balance between introduction and other parts of the manuscript, we added some information to discussion. Please note, that comparison of efficiency of ALA in comparison to other oxidants was not the goal of our study - we have no group of patients treated with other antioxidants as well as appropriate controls. So, we feel that comparison of effectivity of ALA with other antioxidants would be too speculative in the context of our paper.
Comment 8: The authors should describe how DM, especially T2DM connects to ROS excess, insulin resistance, and positive effects of ALA as antioxidant molecules. From the paragraph written, the connection is not discussed well in biological terms. Lines 85-96: It would be important to provide more details on pharmacokinetics and pharmacology of ALA (Lines 87-89) so that the authors’ argument for the use of ALA for less than 10 days could be a practical option (the current argument is too vague).
Response: We agree with these comments and have made necessary corrections. Information regarding pharmacokinetics is added to discussion (in order to keep more balance in proportions of introduction and discussion).
Comment 9: Overall, agree with the use of ALA in the attempt to ameliorate the symptoms associated with DPN, but it is important to have the introduction written with the underlying biological processes being considered. Furthermore, the authors still did not make the distinction between T1DM and T2DM well in the introduction, especially not mentioning T1DM is an autoimmune disease. It is not typically associated with insulin resistance.
Response: Corrected
Comment 10: Line 206: Suggest starting a new paragraph at, “It is interesting to note …” At this point, did the authors decide to examine the same pain parameters or any change in biomarkers studied? It may be important to note these so that it may be possible to determine the most appropriate length of treatment without reducing the effectiveness of ALA therapy. In the discussion, the authors described that It is also important to note how long the pain reduction lasted. This information would be critically important if this administration were to be clinically considered, one needs to know how long the effectiveness lasts.
Response: We made a separate paragraph, as recommended by Reviewer, and added some additional details to this part of discussion.